# Phenome-wide Mendelian randomisation analysis of 378,142 cases reveals risk factors for eight common cancers

Molly Went [1,11] ✉, Amit Sud [1,2,3,4,5,11], Charlie Mills [1,11], Abi Hyde [1,6,11], Richard Culliford[1], Philip Law [1], Jayaram Vijayakrishnan [1], Ines Gockel [7], Carlo Maj [8], Johannes Schumacher [8], Claire Palles [9], Martin Kaiser [1,10] & Richard Houlston [1]

For many cancers there are only a few well-established risk factors. Here, we use summary data from genome-wide association studies (GWAS) in a Mendelian randomisation (MR) phenome-wide association study (PheWAS) to identify potentially causal relationships for over 3,000 traits. Our outcome datasets comprise 378,142 cases across breast, prostate, colorectal, lung, endometrial, oesophageal, renal, and ovarian cancers, as well as 485,715 controls. We complement this analysis by systematically mining the literature space for supporting evidence. In addition to providing supporting evidence for well-established risk factors (smoking, alcohol, obesity, lack of physical activity), we also find sex steroid hormones, plasma lipids, and telomere length as determinants of cancer risk. A number of the molecular factors we identify may prove to be potential biomarkers. Our analysis, which highlights aetiological similarities and differences in common cancers, should aid public health prevention strategies to reduce cancer burden. We provide a R/Shiny app to visualise findings.

Cancer is currently the third major cause of death with an estimated 18.1 million new cases and nearly 10 million cancer deaths in 2020[1]. By 2030 it is predicted there are likely to be 26 million new cancer cases and 17 million cancer-related deaths annually[2]. Such projections have renewed efforts to identify risk factors to inform cancer prevention programmes.

For many cancers, despite significant epidemiological research, there are few well-established risk factors. Although randomised-controlled trials (RCTs) are the gold standard for establishing causal relationships, they are often impractical or unfeasible because of cost, time, and ethical issues. Conversely, case-control studies can be complicated by biases such as reverse causation and confounding. Mendelian randomisation (MR) is an analytical strategy that uses germline genetic variants as instrumental variables (IVs) to infer potentially causal relationships (Fig. 1A)[3]. The random assortment of these genetic variants at conception mitigates against reverse causation bias. Moreover, in the absence of pleiotropy (*i.e.* the presence of an association between variants and disease through additional pathways), MR can provide unconfounded disease risk estimates. Elucidating disease causality using MR is gaining popularity especially given the availability of data from large genome-wide association studies (GWAS) and well-developed analytical frameworks[3].

[1]Division of Genetics and Epidemiology, The Institute of Cancer Research, London, UK. [2]Department of Medical Oncology, Dana-Farber Cancer Institute, Boston, MA, USA. [3]Broad Institute of MIT and Harvard, Cambridge, MA, USA. [4]Harvard Medical School, Boston, MA, USA. [5]Centre for Immuno-Oncology, Nuffield Department of Medicine, University of Oxford, Oxford, UK. [6]Department of Engineering, University of Cambridge, Cambridge, UK. [7]Department of Visceral, Transplant, Thoracic and Vascular Surgery, University Hospital of Leipzig, Leipzig, Germany. [8]Center for Human Genetics, University Hospital of Marburg, Marburg, Germany. [9]Institute of Cancer and Genomic Sciences, University of Birmingham, Birmingham, UK. [10]The Royal Marsden Hospital NHS Foundation Trust, London, UK. [11]These authors contributed equally: Molly Went, Amit Sud, Charlie Mills, Abi Hyde. ✉e-mail: molly.went@icr.ac.uk

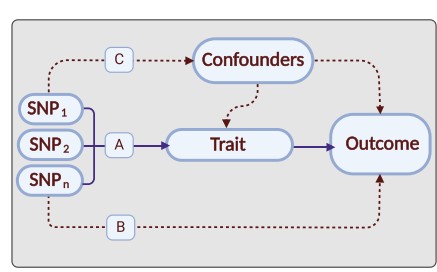

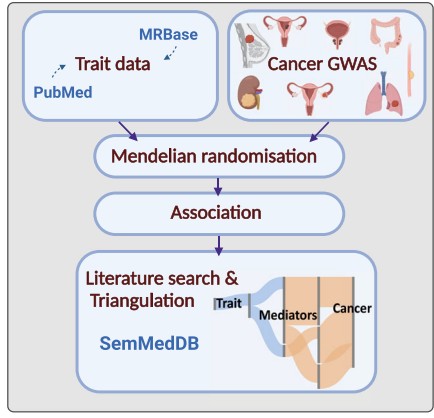

A                                               B

**Fig. 1 | Principles of Mendelian randomisation (MR) and study overview.
A Assumptions in MR that need to be satisfied to derive unbiased causal effect
estimates.** Dashed lines represent direct causal and potential pleiotropic effects
that would violate MR assumptions. A, indicates genetic variants used as IVs are
strongly associated with the trait; B, indicates genetic variants only influence cancer
risk through the trait; C, indicates genetic variants are not associated with any
measured or unmeasured confounders of the trait-cancer relationship. SNP, single-
nucleotide polymorphism; **B Study overview**. Genetic variants serving as instru-
ments for exposure traits under investigation were identified from MRBase or
PubMed. GWAS data for the eight cancers was acquired and MR analysis was per-
formed. Results were triangulated through literature mining to provide supporting
evidence for potentially causal relationships. Created with BioRender.com. GWAS
genome-wide association study.

Most MR studies of cancer have been predicated on assumptions
about disease aetiology or have sought to evaluate purported asso-
ciations from conventional observational epidemiology[3,4]. A recently
proposed agnostic strategy, termed MR-PheWAS, integrates the
phenome-wide association study (PheWAS) with MR methodology to
identify potential causal relationships considering hitherto previously
unexamined traits[5].

To identify potentially causal relationships for eight common
cancers: breast, prostate, colorectal (CRC), lung, endometrial, oeso-
phageal, renal cell carcinoma (RCC), ovarian, and reveal intermediates
of risk, we conducted a MR-PheWAS study utilising 378,142 cases and
485,715 controls. We integrated findings with a systematic mining of
the literature space to provide supporting evidence and derive a more
comprehensive description of disease aetiology (Fig. 1B)[6].

## Results

### Phenotypes and genetic instruments
After filtering we analysed 3661 traits, proxied by 336,191 genetic
variants in conjunction with summary genetic data from published
GWAS of breast, prostate, colorectal, lung, endometrial, oesophageal,
renal, and ovarian cancers (Table 1; Supplementary Data 1). The num-
ber of single nucleotide polymorphisms (SNPs) used as genetic
instruments for each trait ranged from one to 1335. Figure 2 shows the
power of our MR study to identify potentially causal relationships
between each of the genetically defined traits and each cancer type.
The median proportion of variance explained (PVE) by SNPs used as IVs
for each of the 3,661 traits evaluated as risk factors was 3.4%
(0.01–84%). Our power to demonstrate relationships a priori for each
cancer type reflects in part inevitably the size of respective GWAS
datasets (Supplementary Data 2).

### Causal associations predicted by MR
To aid interpretation, we grouped traits related to established
cancer risk factors (*i.e.* smoking, obesity and alcohol) and those
for which current evidence is inconclusive into the following
categories, using a similar approach to Markozannes et al.[4]: cardi-
ometabolic; dietary intake; anthropometrics; immune and inflam-
matory factors; fatty acid (FA) and lipoprotein metabolism;
lifestyle, reproduction, education and behaviour; metabolomics
and proteomics; miscellaneous.

Given the large number of traits being evaluated, we categorised
the support for potentially causal relationships between non-binary
traits and cancers into four hierarchical levels of statistical significance
a priori: robust, probable, suggestive, and non-significant (Fig. 3;
Methods). Out of the 27,066 graded associations, MR analyses pro-
vided robust evidence for a potentially causal relationship with 123
phenotypes (0.5% of total MR analyses), 174 with probable evidence
(0.6% of total), 1652 with suggestive evidence (6% of total). Across the
eight cancer types, the largest number of robust associations were
observed for endometrial cancer with 37 robust associations, followed
by RCC ($n = 32$), CRC ($n = 21$), lung ($n = 20$), breast ($n = 10$), oesopha-
geal ($n = 3$) and prostate cancer ($n = 1$). No robust MR associations were
observed for ovarian cancer (Supplementary Data 3).

Across all the cancer types, anthropometric traits showed the
highest number of robust relationships ($n = 32$; 0.1%), followed by
lifestyle, reproduction, education, and behaviour ($n = 17$; 0.06%). No
robust associations were observed for dietary intake or cardiometa-
bolic categories (Supplementary Data 3).

To visualise the strength and direction of effect of the relationship
between each of the traits examined and risk of each cancer type and,
where appropriate, their respective subtypes we provide a R/Shiny app
(https://software.icr.ac.uk/app/mrcan). Figure 4 shows a screenshot of
the app for selected traits across the eight different types of cancer.

Many of the identified potentially causal relationships, especially
those that were statistically robust or probable, have been reported in
previous MR studies and are related to established risk factor
categories[4,7,8]. Notably: (i) the relationship between metrics of
increased body mass index (BMI) with an increased risk of colorectal
(*Robust*, $OR_{SD} = 1.19$, 95% CI: 1.11–1.27, $P = 2.01 \times 10^{-7}$), lung (*Suggestive*,
$OR_{SD} = 1.22$, 95% CI: 1.11–1.34, $P = 3.25 \times 10^{-5}$), renal (*Robust*, $OR_{SD} = 1.63$,
95% CI: 1.44–1.85, $P = 2.19 \times 10^{-14}$), endometrial (*Robust*, $OR_{SD} = 1.90$,
95% CI: 1.67–2.15, $P = 3.92 \times 10^{-23}$) and ovarian (*Suggestive*, $OR_{SD} = 1.11$,
95% CI: 1.01–1.22, $P = 2.98 \times 10^{-2}$) cancers[9]; (ii) cigarette smoking with
an increased risk of lung cancer[10]; (iii) traits related to higher alcohol
consumption and increased risk of oesophageal (*Suggestive*,
$OR_{SD} = 2.69$, 95% CI: 1.58–4.49, $P = 2.76 \times 10^{-4}$), CRC (*Suggestive*,
$OR_{SD} = 1.39$, 95% CI: 1.01–1.91, $P = 4.53 \times 10^{-2}$), lung (*Probable*,
$OR_{SD} = 1.55$, 95% CI: 1.18–2.04, $P = 1.49 \times 10^{-3}$), RCC (*Suggestive*,
$OR_{SD} = 1.25$, 95% CI: 1.03 - 1.53, $P = 2.42 \times 10^{-2}$), endometrial (*Suggestive*,
$OR_{SD} = 1.23$, 95% CI: 1.01–1.8515, $P = 4.41 \times 10^{-2}$) and ovarian (*Suggestive*,

## Table 1 | Details of cancer genome-wide association studies used in the Mendelian randomisation analysis

| Cancer | Cases | Controls | PubMed ID | No. of contributing studies | GWAS catalogue ID |
|---|---|---|---|---|---|
| Breast | 133384 | 113789 | 32424353 | 82 | GCST010098 GCST010099 GCST010100 |
| Breast triple negative | 2006 | 20815 | | | |
| Breast luminal A | 7325 | | | | |
| Breast luminal B | 1682 | | | | |
| Breast HER2 enriched | 718 | | | | |
| Breast HER2 negative luminal B | 1779 | | | | |
| Colorectal | 73673 | 86854 | 36539618 | 16 | GCST90129505 |
| Endometrial | 8758 | 46126 | 30093612 | 17 | GCST006465 |
| Lung | 29266 | 56450 | 28604730 | 26 | GCST004744 GCST004746 GCST004747 GCST004748 GCST004749 GCST004750 |
| Lung ever-smoked | 23223 | 16964 | | | |
| Lung never-smoked | 2355 | 7504 | | | |
| Lung adenocarcinoma | 11273 | 55483 | | | |
| Lung squamous cell carcinoma | 7426 | 55627 | | | |
| Lung small cell lung cancer | 2664 | 21444 | | | |
| Oesophageal | 16790 | 32476 | 35882562 | 5 | NA |
| Ovarian | 26293 | 68502 | 28346442 | 77 | GCST004415 GCST004416 GCST004417 GCST004418 GCST004419 GCST004461 GCST004462 GCST004478 GCST004479 GCST004480 GCST004481 |
| Ovarian invasive high grade serous | 13037 | 40941 | | | |
| Ovarian all serous | 16003 | 40941 | | | |
| Ovarian invasive mucinous | 1417 | 40941 | | | |
| Ovarian all mucinous | 2566 | 40941 | | | |
| Ovarian all low malignant potential | 3103 | 40941 | | | |
| Ovarian invasive low grade serous and low malignant potential serous | 2966 | 40941 | | | |
| Ovarian invasive low grade serous cases | 1012 | 40941 | | | |
| Ovarian endometrioid | 2810 | 40941 | | | |
| Ovarian clear cell | 1366 | 40941 | | | |
| Ovarian low malignant potential serous | 1954 | 40941 | | | |
| Ovarian low malignant potential mucinous | 1149 | 40941 | | | |
| Prostate | 79194 | 61112 | 29892016 | 8 | GCST006085 |
| Renal | 10784 | 20406 | 28598434 | 5 | GCST004710 |

The number of cases and controls, the number of studies contributing to the meta-analyses and the associated publication and GWAS catalogue IDs are provided for each cancer GWAS. Where applicable, the number of cases and controls in given histological subtypes are also provided.

$OR_{SD} = 1.22$, 95% CI: 1.05–1.40, $P = 7.32 \times 10^{-3}$) cancers[11]; (iv) traits indicative of reduced physical activity and sedentary behaviour with an increased risk of multiple cancers, including breast, lung, colorectal and endometrial[12]. As anticipated, exposure traits pertaining to cigarette smoking were not causally related to lung cancer in never smokers. Paradoxically, but as reported in previous MR analyses, increased BMI was associated with reduced risk of prostate (*Suggestive*, $OR_{SD} = 0.82$, 95% CI: 0.70–0.95, $P = 1.03 \times 10^{-2}$) and breast (*Probable*, $OR_{SD} = 0.84$, 95% CI: 0.76–0.93, $P = 8.40 \times 10^{-4}$) cancer, and an inverse relationship between smoking and prostate cancer risk was shown[9,13]. Our analysis also supports the reported relationship between higher levels of sex hormone-binding globulin with reduced endometrial cancer risk (*Robust*, $OR_{SD} = 0.81$, 95% CI: 0.74–0.89, $P = 9.00 \times 10^{-6}$) and a relationship between testosterone with risk of endometrial (*Probable*, $OR_{SD} = 1.48$, 95% CI: 1.12–1.96, $P = 5.32 \times 10^{-3}$) and breast (*Probable*, $OR_{SD} = 1.24$, 95% CI: 1.09–1.42, $P = 1.43 \times 10^{-3}$) cancer[14,15]. Notably, exposure traits related to testosterone levels were only predicted to be causally associated with luminal-A and luminal-B breast cancer subtypes.

We found associations between genetically predicted high serum vitamin B12 with increased risks of CRC (*Suggestive*, $OR_{SD} = 1.09$, 95% CI: 1.01–1.18, $P = 2.53 \times 10^{-2}$) and prostate (*Suggestive*, $OR_{SD} = 1.08$, 95% CI: 1.02–1.14, $P = 8.87 \times 10^{-3}$) cancer, higher serum calcium (*Suggestive*, $OR_{SD} = 1.19$, 95% CI: 1.05–1.35, $P = 5.92 \times 10^{-3}$) and 25-hydroxyvitamin-D (*Suggestive*, $OR_{SD} = 1.18$, 95% CI: 1.00–1.38, $P = 4.63 \times 10^{-2}$) with an increased risk of RCC, higher blood selenium with decreased risks of CRC (*Suggestive*, $OR_{SD} = 0.91$, 95% CI: 0.85–0.98, $P = 9.49 \times 10^{-3}$) and oesophageal (*Suggestive*, $OR_{SD} = 0.84$, 95% CI: 0.72–0.99, $P = 3.42 \times 10^{-2}$) cancer and higher methionine (*Suggestive*, $OR_{SD} = 0.09$, 95% CI: 0.01–0.99, $P = 4.90 \times 10^{-2}$) and zinc (*Suggestive*, $OR_{SD} = 0.94$, 95% CI: 0.89–0.99, $P = 1.77 \times 10^{-2}$) with reduced CRC risk. We observed no association between genetically predicted blood levels of circulating carotenoids or vitamins B6 and E for any of the cancers. With respect to dietary intake our analysis demonstrated associations between genetically predicted higher levels of coffee intake (*Probable*, $OR_{SD} = 0.67$, 95% CI: 0.55–0.82, $P = 1.03 \times 10^{-4}$), oily fish (*Probable*, $OR_{SD} = 0.66$, 95% CI: 0.52–0.84, $P = 5.41 \times 10^{-4}$), and cheese intake (*Probable*, $OR_{SD} = 0.75$, 95% CI: 0.64–0.89, $P = 1.08 \times 10^{-3}$) with reduced CRC risk and associations between genetically predicted beef (*Suggestive*, $OR_{SD} = 1.65$, 95% CI: 1.05–2.60, $P = 3.07 \times 10^{-2}$) and poultry (*Suggestive*, $OR_{SD} = 2.10$, 95% CI: 1.06–4.16, $P = 3.24 \times 10^{-2}$) intake and elevated CRC risk.

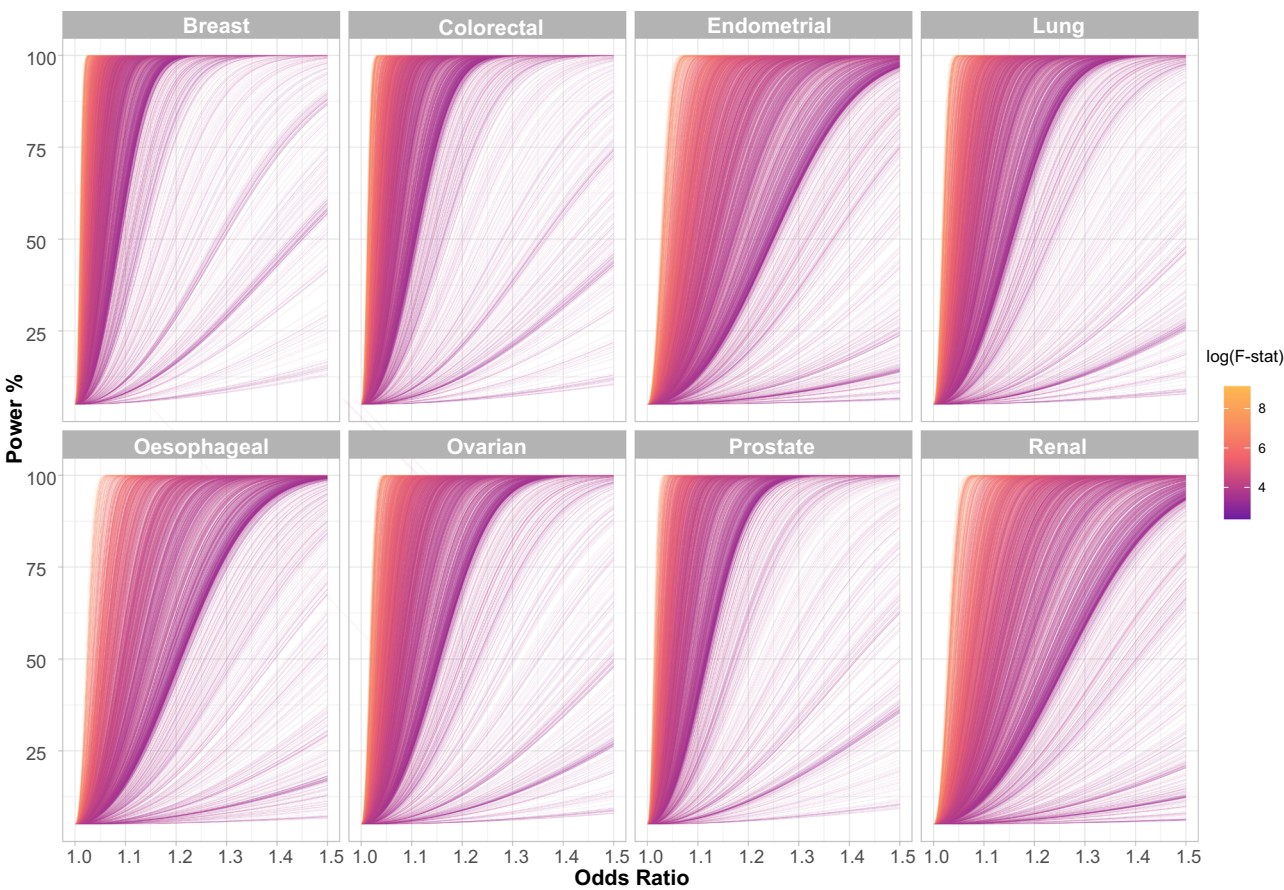

**Fig. 2 | Power to predict causal relationships in the Mendelian randomisation analysis across the eight different cancers.** Each line represents an individual trait with the line colour indicating the F-statistic, a measure of instrument strength. The analysis of most traits is well powered across a modest range of odds ratios. Generally, better powered traits are those with a higher F-statistic. F-stat: F-statistic.

In terms of glucose homeostasis, no relationship between genetically predicted blood glucose or glycated haemoglobin was shown for any of the eight cancers. However, higher levels of genetically predicted levels of fasting insulin (*Probable*, $OR_{SD} = 1.78$, 95% CI: 1.25–2.52, $P = 1.33 \times 10^{-3}$) and insulin growth factor 1 (IGF-1) (*Suggestive*, $OR_{SD} = 1.06$, 95% CI: 1.01–1.12, $P = 3.26 \times 10^{-2}$) and lower proinsulin (*Probable*, $OR_{SD} = 0.89$, 95% CI: 0.82 - 0.96, $P = 3.09 \times 10^{-3}$) showed associations with CRC. Additionally, an association between proinsulin and RCC (*Suggestive*, $OR_{SD} = 0.80$, 95% CI: 0.67–0.96, $P = 1.50 \times 10^{-2}$), fasting insulin and lung (*Suggestive*, $OR_{SD} = 1.40$, 95% CI: 1.03 - 1.90, $P = 3.29 \times 10^{-2}$) and endometrial (*Suggestive*, $OR_{SD} = 1.76$, 95% CI: 1.02–3.03, $P = 4.24 \times 10^{-2}$) cancers, and IGF-1 levels and breast cancer (*Probable*, $OR_{SD} = 1.07$, 95% CI: 1.02–1.13, $P = 6.21 \times 10^{-3}$) was observed.

Amongst genetically predicted higher levels of lipoproteins, the only associations were between high density lipoprotein cholesterol (HDL-C) and breast cancer risk (*Probable*, $OR_{SD} = 1.08$, 95% CI: 1.03–1.12, $P = 6.28 \times 10^{-4}$), low density lipoprotein cholesterol (LDL-C) an elevated risk of CRC (*Suggestive*, $OR_{SD} = 1.10$, 95% CI: 1.01–1.20, $P = 2.18 \times 10^{-2}$), and total cholesterol and increasing ovarian cancer risk (*Suggestive*, $OR_{SD} = 1.05$, 95% CI: 1.01–1.09, $P = 2.67 \times 10^{-2}$). Genetically predicted levels of plasma FAs showed an association with reduced cancer risk. Specifically, for the omega-6 polyunsaturated FAs, increased levels of arachidonic acid (20:4n6) (*Suggestive*, $OR_{SD} = 1.04$, 95% CI: 1.02–1.05, $P = 6.11 \times 10^{-5}$) and gamma-linoleic acid (18:3n6) (*Suggestive*, $OR_{SD} = 35.29$, 95% CI: 13.65–91.24, $P = 1.94 \times 10^{-13}$) and lower levels of linoleic acid (18:2n6) (*Suggestive*, $OR_{SD} = 0.96$, 95% CI: 0.95–0.97, $P = 3.11 \times 10^{-13}$) and adrenic acid (22:4n6) (*Suggestive*, $OR_{SD} = 3.28$, 95% CI: 2.34–4.59, $P = 5.88 \times 10^{-12}$) with increased risk of CRC; for the omega-3 polyunsaturated FAs, linoleic acid (*Suggestive*,

$OR_{SD} = 1.02$, 95% CI: 1.00–1.04, $P = 3.05 \times 10^{-2}$) and eicosapentaenoic acid (*Suggestive*, $OR_{SD} = 0.42$, 95% CI: 0.19–0.94, $P = 3.44 \times 10^{-2}$) showed an association with ovarian cancer risk while arachidonic acid was associated with endometrial cancer (*Suggestive*, $OR_{SD} = 0.98$, 95% CI: 0.97–0.99, $P = 2.83 \times 10^{-3}$). Performing a leave-one-out and single SNP analysis (Supplementary Data 4 and 5, respectively) we found, similar to previously published work, that the majority of associations with respect to omega-3 and omega-6 fatty acids are driven by correlated associations within the *FADS* locus[16,17].

A relationship between longer lymphocyte telomere length (LTL) and an increased risk of six of the eight cancer types was identified - RCC (*Robust*, $OR_{SD} = 2.01$, 95% CI: 1.65–2.45, $P = 3.27 \times 10^{-12}$), lung (*Robust*, $OR_{SD} = 1.61$, 95% CI: 1.41–1.84, $P = 2.48 \times 10^{-12}$), breast (*Probable*, $OR_{SD} = 1.12$, 95% CI: 1.04–1.20, $P = 2.07 \times 10^{-3}$), prostate (*Probable*, $OR_{SD} = 1.25$, 95% CI: 1.10–1.43, $P = 9.77 \times 10^{-4}$), colorectal (*Suggestive*, $OR_{SD} = 1.13$, 95% CI: 1.00–1.28, $P = 4.24 \times 10^{-2}$) and ovarian cancer (*Suggestive*, $OR_{SD} = 1.18$, 95% CI: 1.05–1.33, $P = 4.88 \times 10^{-3}$).

In addition to a robust association between higher HLA-DR dendritic plasmacytoid levels and risk of prostate cancer ($OR_{SD} = 1.05$, 95% CI: 1.03–1.06, $P = 5.22 \times 10^{-10}$), 26 probable associations between genetically predicted levels of other circulating immune and inflammatory factors were shown across the cancers studied. These included higher levels of IL-18 with reduced risk of lung cancer (*Probable*, $OR_{SD} = 0.89$, 95% CI: 0.83–0.96, $P = 2.00 \times 10^{-3}$), with specificity for lung cancer in never smokers. For proteomic traits, we conducted a Bayesian colocalisation analysis to determine whether genetic variants influencing protein levels and cancer risk are shared by considering the strongest proteomic associations with a clear gene target and a cis-IV (*i.e.* within 1 Mb; **Methods**) with $P$-value $< 1 \times 10^{-6}$ in the outcome

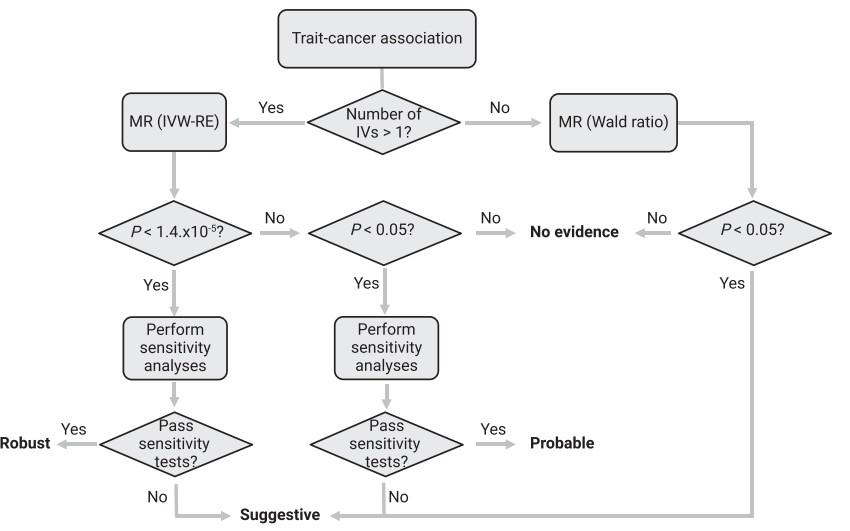

**Sensitivity tests**

- **Weighted median estimate**: performs MR and remains accurate in the presence of weak IVs.
- **Mode based estimate**: performs MR and remains accurate in the presence of weak IVs.
- **MR-Egger**: tests for directional pleiotropy.
- **MR-Steiger**: provides an estimate of the direction of effect.

**Fig. 3 | Hierarchical classification of associations.** Potentially causal relationships between non-binary traits and cancers were categorised into four hierarchical levels of statistical significance a priori; robust ($P_{IVW-RE} < 1.4 \times 10^{-5}$; corresponding to a $P$-value of 0.05 after Bonferroni correction for multiple testing (0.05/3,500), $P_{WME}$ or $P_{MBE} < 0.05$, predicted true causal direction and >1 IVs), probable ($P_{IVW-RE} < 0.05$, $P_{WME}$ or $P_{MBE} < 0.05$, predicted true causal direction and >1 IVs), suggestive ($P_{IVW-RE} < 0.05$ or $P_{WALD} < 0.05$), and non-significant ($P_{IVW-RE} \geq 0.05$ or $P_{WALD} \geq 0.05$). Weighted median estimates (WME)[52] and mode-based estimates (MBE)[53] were used in addition to an inverse weighted random effects (IVW-RE) model, to assess the robustness of our findings, while MR-Egger regression assessed the extent to which directional pleiotropy could affect causal estimates[54]. MR-Steiger was used to ascertain that the exposure trait influenced the outcome and not vice versa[55]. Binary traits were classified associations as being supported ($P < 0.05$) or not supported ($P > 0.05$). MR, Mendelian randomisation; IV, instrumental variable.

cancer. We identified KDEL motif-containing protein 2 (*KDELC2*) and RCC, as well as Copine-1 (*CPNE1*) and Immunoglobulin superfamily containing leucine-rich repeat protein 2 (*ISLR2*) and breast cancer as having a high posterior probability of a shared variant (*i.e.* PP_H4 > 0.8). In contrast, Kunitz-type protease inhibitor 2 (*SPINT2*) and prostate cancer, as well as Semaphorin-3G (*SEMA3G*) and CRC, were shown to have distinct variants at the gene target (*i.e.* PP_H3 > 0.8; Supplementary Data 6). Results for the IV at Histo-blood group ABO system transferase (*ABO*) with ovarian cancer were indeterminate (PP_H4 = 0.67 and PP_H3 = 0.33).

Our MR analysis provides support for a relationship between rectal polyps and CRC (β = 95.59, Standard Error (SE) = 4.99, $P = 6.88 \times 10^{-82}$)[18], benign breast disease and breast cancer[19], and oesophageal reflux with risk of oesophageal cancer (β = 0.27, SE = 0.08, $P = 1.30 \times 10^{-3}$) (Supplementary Data 7)[20]. Other associations included possible relationships between pulmonary fibrosis and lung cancer[21], as well as the relationship between a diagnosis of schizophrenia and lung cancer (β = 0.10, SE = 0.04, $P = 2.89 \times 10^{-2}$), which has been previously reported in conventional epidemiological studies[22]. It was noteworthy, however, that we did not find evidence to support the purported relationship between hypertension and risk of developing RCC[23]. Similarly, our analysis did not provide evidence to support a causal relationship between either type 1 or type 2 diabetes and an increased cancer risk.

## Multivariable MR of biologically related traits

Selected traits within our analysis may show pleiotropic effects with other traits and work by Burgess et al. [24] has shown that MR can only assess the causal effect of a risk factor on an outcome by using genetic variants that are solely associated with the risk factor of interest. To address pleiotropy we performed multivariable MR (MVMR) as a form of mediation analysis focusing on known biologically related traits. Specifically, we examined the role of IGF-1 and height on breast and

colorectal cancer risk[25]; lipid traits on breast and colorectal cancer risk[26,27]; and fasting insulin, sex hormone-binding globulin levels (SHBG), BMI and testosterone on endometrial cancer risk[28] (Supplementary Data 8). In the MVMR analysis of HDL-C, LDL-C and triglyceride levels, we found the relationship of increasing HDL cholesterol with breast cancer risk and increasing LDL-C with colorectal cancer risk remained significant in a model accounting for these biologically related traits (OR_MVMR = 1.06, $P_{MVMR} = 0.03$ and OR_MVMR = 1.09, $P_{MVMR} = 0.04$, respectively). Considering height and IGF-1 and their association with CRC risk and breast cancer risk, IGF-1 remained significantly associated with breast cancer risk (OR_MVMR = 1.06, $P_{MVMR} = 0.049$), while height remained significantly associated with colorectal cancer risk (OR_MVMR = 1.06, $P_{MVMR} = 0.045$). In contrast IGF-1 became non-significant ($P = 0.16$), which may suggest that the relationship between IGF-1 levels and CRC is mediated through the relationship with height. Finally, MVMR of fasting insulin, SHBG, BMI and testosterone and their effect on endometrial cancer, attenuated the significance of association ($P > 0.5$) of fasting insulin and bioavailable testosterone with the outcome, while SHBG and BMI remained significant, but with a modest decrease in effect size (OR_MVMR = 0.61, $P_{MVMR} = 0.02$ and OR_MVMR = 1.65, $P_{MVMR} = 6.37 \times 10^{-5}$). Hence this suggests that bioavailable testosterone and fasting insulin do not have an independent effect on endometrial cancer risk and the associations are likely to be mediated, at least in part, through SHBG and BMI.

## Literature-mined support for MR defined relationships

To provide support for the associations and to gain molecular insights into the underlying biological basis of relationships we performed triangulation through systematic literature mining. We identified 55,105 literature triples across the eight different cancer types and 680,375 literature triples across the MR defined putative risk factors (Supplementary Data 9). Overlapping risk factor-cancer pairings from our MR analysis yielded on average 49 potential causal relationships.

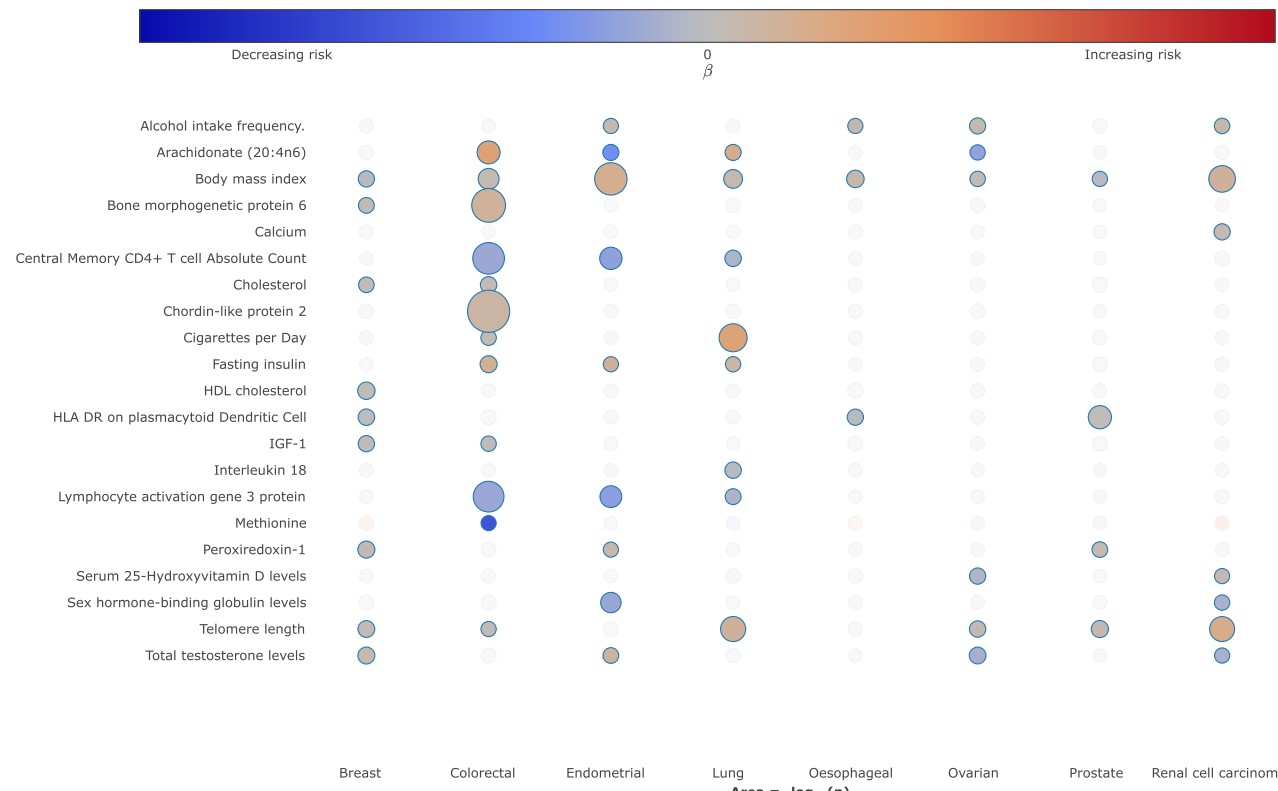

**Fig. 4 | Bubble plot of the potentially causal relationship between selected traits and risk of different cancers.** The columns correspond to different cancer types. The colours on the heatmap correspond to the strength of associations (odds ratio) and their direction (red positively correlated, blue negatively correlated). *P*-values represent the results from two-sided tests and are unadjusted. The size of each node corresponds to the -log₁₀ *P*-value, with increasing size indicating a smaller *P*-value. In the available R/Shiny app (https://software.icr.ac.uk/app/mrcan), moving the cursor on top of each bubble will reveal the underlying MR statistics.

Supplementary Data 10 stratifies the literature space size by trait category while recognising that identified relationships with a small literature space could be reflective of deficiencies in semantic mapping relationships with large literature spaces supporting triangulation. Supplementary Data 11 provides the complete list of potential mediators for each trait. Illustrating the use of triangulation using a large literature space (defined herein as >50 triples) to support potentially causal relationships, Fig. 5 highlights four notable examples (IGF-1, LAG-3, IL-18, and PRDX1).

IGF-1, which is reported to play a role in multiple cancers, appears to mediate its effect in part through beta-catenin and BRAF signalling, modulating CRC and breast cancer risk[29]. Whilst LAG-3 inhibition is an attractive therapeutic target in restoring T-cell function, we demonstrate genetically elevated LAG-3 levels as being associated with reduced CRC, endometrial and lung cancer. In all three of these cancers, the association appears to be at least partly mediated through IL-10. The seemingly paradoxical relationship between LAG-3 levels and tumourgenesis may reflect potentiation of T-cell function by serum LAG-3 rather than cell membrane expressed LAG-3[30]. We identify genetically predicted IL-18 levels as being associated with an increased risk of lung cancer. Our literature mining also supports a role for the decoy inhibitory protein, IL-18BP as being a mediator of lung cancer risk as well as IL-10, IL-12, IL-4 and TNF[31]. Finally, PRDX1, a member of the peroxiredoxin family of antioxidant enzymes, interacts with the androgen receptor to enhance its transactivation resulting in increased EGFR-mediated signalling and an increased prostate cancer risk[32].

## Discussion

By performing a MR-PheWAS we have been able to agnostically examine the relationship between multiple traits and the risk of eight different cancer types, restricted only by the availability of suitable genetic instruments. Importantly, many of the traits we examined have not previously been the subject of conventional epidemiological studies or been assessed by MR. Comparing our work with a recent systematic review of the previously published MR studies of cancer, less than 10% of the MR exposures in this study had been the subject of previous investigations[4]. In addition, 85% of those traits which we found were significant had not previously been examined. Even for risk factors that were examined in many previous analyses, the number of cases and controls in our study has afforded greater power to identify potential causal associations. This has allowed us to exclude large effects on cancer risk for most exposure traits examined.

In addition to predicting causal relationships for the well-established lifestyle traits, which validates our approach, we implicate other lifestyle factors that have been putatively associated by observational epidemiology contributing to cancer risk. For example, the protective effects of physical activity (*Suggestive*) with lung cancer risk, oily fish (*Probable*) for CRC risk and fresh/dried fruit intake (*Probable*) for breast cancer risk. Several of the potentially causal relationships we identify have been the subject of studies of individual traits and include the association between longer LTL with increased risk of RCC and lung cancers (*Robust*); sex steroid hormones and risk of breast and endometrial cancer and circulating lipids with CRC and breast cancer. Clustering of MR predicted causal effect sizes for each trait cancer relationship highlights the importance of risk factors common to many cancers but also reveal differences in their impact in part likely to be reflective of underlying biology (Fig. 6).

Using genetic instruments for plasma proteome constituents has allowed us to identify hitherto unexplored potential risk factors

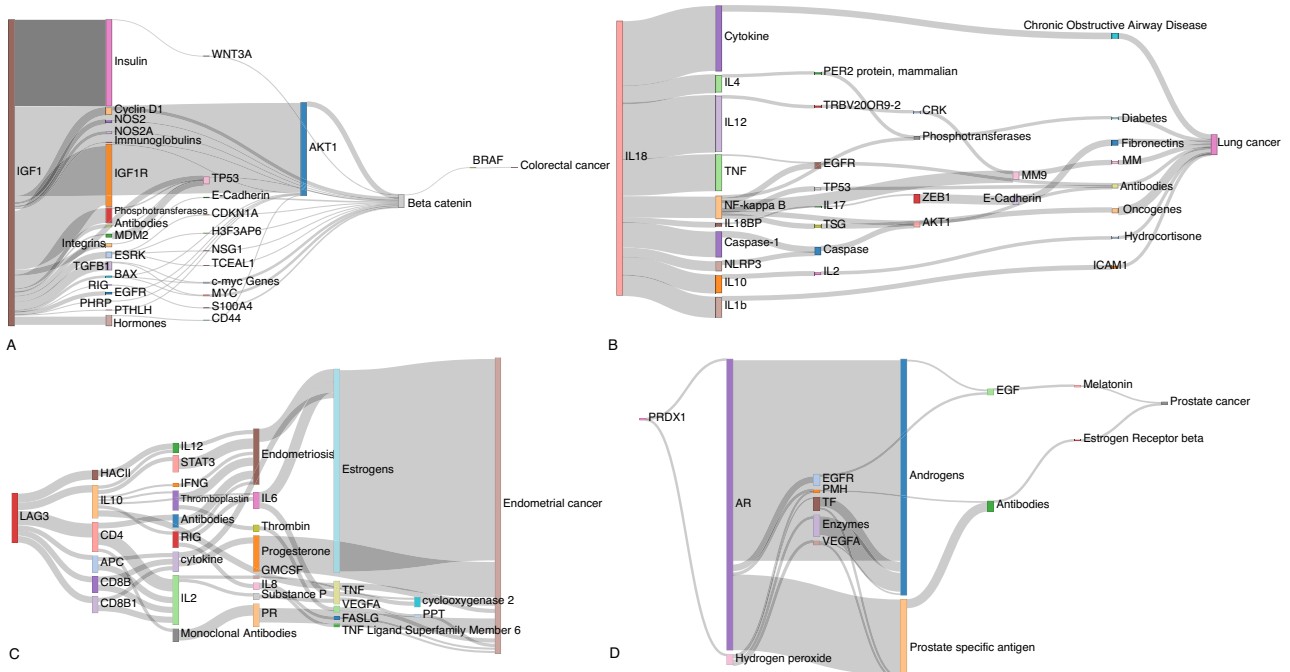

**Fig. 5 | Sankey diagram of literature spaces for exemplar cancer risk factors.** These diagrams illustrate the relationship between exposure traits and cancers via their linked literature triples. The thickness of the line connecting two mediating traits indicates the frequency with which that triple is mentioned in the literature. Relationships for: **A** *IGF-1* and colorectal cancer; **B** *IL-18* and lung cancer; **C** *LAG–3* and endometrial cancer; **D** *PRDX1* and prostate cancer. AR androgen receptor, EGF: epidermal growth factor, EGFR epidermal growth factor receptor, ESRK extracellular signal regulated kinases, GMCSF granulocyte-macrophage colony-stimulating factor, HACII histocompatibility antigens class II, IFNG interferon gamma, MM matrix metalloproteinases, MM9 matrix metalloproteinase 9, PHRP parathyroid hormone-related protein, PMH phosphoric monoester hydrolases, PPT: phenylpyruvate tautomerase, PR progesterone receptor, RIG recombinant interferon-gamma, TF transcription factor, TNF tumour necrosis factor, TSG tumour suppressor genes, VEGFA vascular endothelial growth factor A.

for a number of the cancers, including: the cytokine like molecule, FAM3D, which plays a role in host defence against inflammation associated carcinogenesis with lung cancer[33]; the autophagy associated cytokine cardiotrophin-1 with lung (*Probable*), endometrial (*Suggestive*), prostate (*Suggestive*) and breast (*Suggestive*) cancer and the tumour progression associated antigen CD63 with endometrial cancer[34,35]. Levels of these and other plasma proteins potentially represent biomarkers worthy of future prospective studies. Furthermore, for proteomic traits with *cis*-IVs previous work has found that an MR association with colocalization evidence is associated with a higher likelihood of a particular target-indication pair being successful in drug discovery[36].

A principal assumption in MR is that variants used as IVs are associated with the exposure trait under investigation. We therefore used SNPs associated with exposure traits at genome-wide significance. Furthermore, only IVs from European populations were used to limit bias from population stratification. Our MR analysis does, however, have limitations. Firstly, we were limited to studying phenotypes with genetic instruments available, moreover traits such as food intake or television watching can be highly correlated with other exposures making deconvolution of the causal risk factor problematic[37–39]. While MVMR can be used to account for the correlation between traits, calculation of conditional F-statistics for dietary traits yielded weak instruments (F < 3), which precludes their inclusion in an MVMR model due to weak instrument bias. Secondly, correcting for multiple testing guards against false positives especially when based on a single exposure outcome. However, the potential for false negatives is not unsubstantial. Since we have not adjusted for between trait correlations, our associations are inevitably conservative. Thirdly, for several traits, we had limited power to demonstrate associations of small effect. Fourthly, not unique to our

MR analysis, is the inability of our study to deconvolute time-varying effects of genetic variants as evidenced by the relationship between obesity and breast cancer risk[40]. Finally, as with all MR studies, excluding pleiotropic IVs is challenging. To address this, we incorporated information from weighted median and mode-based estimate methods, to classify the strength of potentially causal associations. For groups of traits susceptible to pleiotropy (*e.g.*, lipids) we also demonstrated how their incorporation into a MVMR model can affect the relationship between these traits and outcome. There are inevitably limitations to such modelling as exemplified by the strong relationship between plasma FA and risk of CRC which has been shown to be driven by the pleiotropic *FADS* locus which has a profound effect on the metabolism of multiple FA through its gene expression[17].

A major concern articulated regarding any MR-PheWAS is the need to provide supporting evidence from alternative sources. Herein we have sought to address this by conducting a systematic interrogation of the literature space and potentially identify intermediates to explain relationships. Furthermore, we performed MVMR to deconvolute relationships where multiple traits appear to influence cancer risk. Although literature mined data can be noisy and driven by publication bias, we have been able to provide a narrative of the potentially causal relationships for several risk factors, which are attractive candidates for molecular validation.

While complementary studies are required to delineate the exact biological mechanisms underpinning associations, our analysis does however highlight important targets for primary prevention of cancer in the population. The limited power to robustly characterise relationships between some exposure traits and cancer in this study, provides an impetus for larger MR studies. Finally, we recognise that MR is not infallible and replication and triangulation of findings using

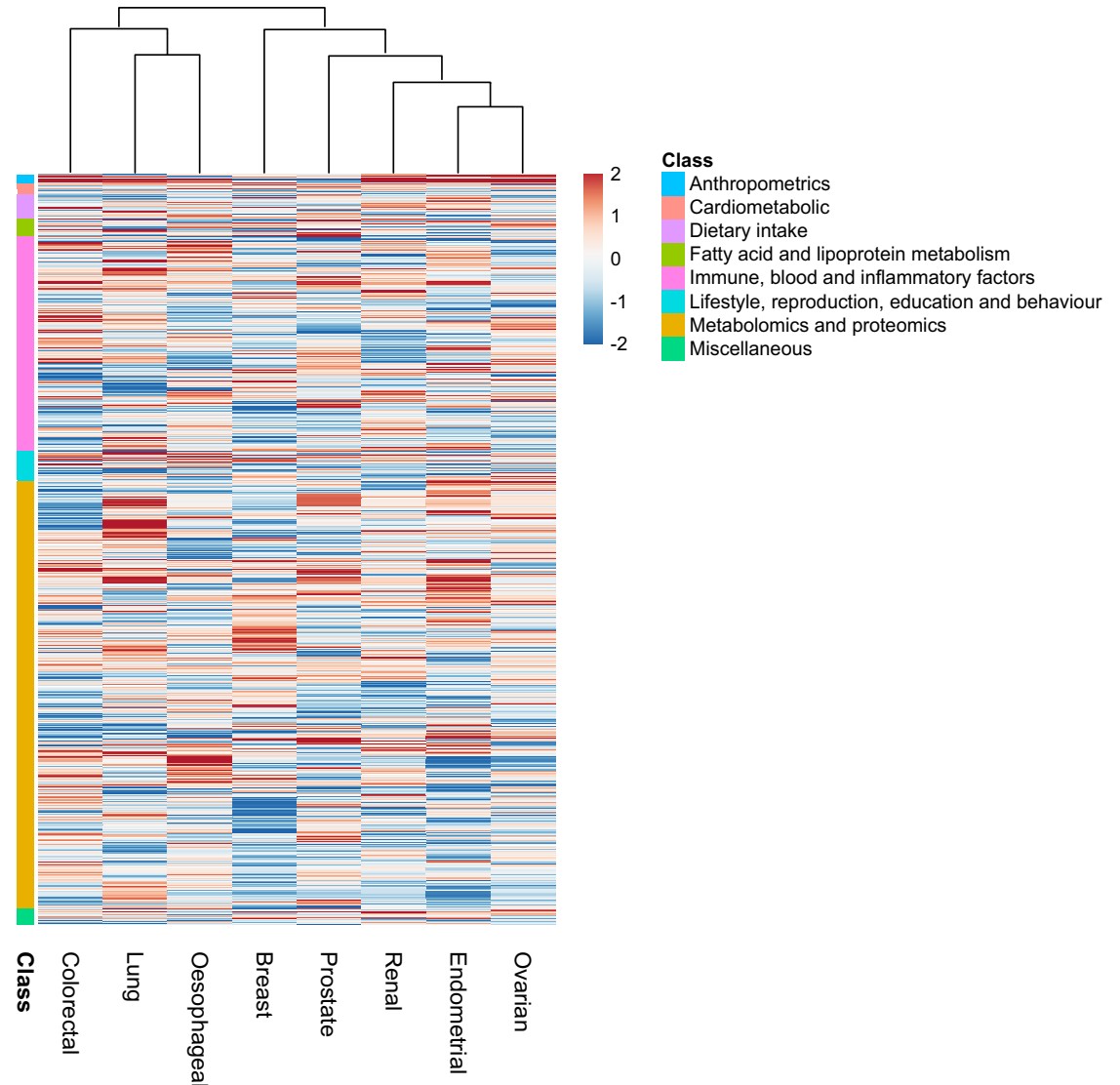

**Fig. 6 | Heatmap and dendrogram showing clustering of potentially causal associations between traits and cancer risk.** Heatmap based on *Z*-statistics using the clustering method implemented in the pheatmap function within R. Colours correspond to the strength of associations and their direction (red positive association with risk, blue inverse association with risk). Trait classes are annotated on the left. Only traits showing an association for at least one cancer type are shown. Further heatmaps for individual classes of traits are shown in Supplementary Figs. 1–8.

different data sources, and if possible, benchmarking against RCTs is highly desirable. Such efforts could identify additional factors as targets to reduce the overall burden of cancer.

## Methods

### Study design
Our study had four elements. Firstly, the identification of genetic variants serving as instruments for exposure traits under investigation; secondly, the acquisition of GWAS data for the eight cancers; thirdly, MR analysis; fourthly, triangulation through literature mining to provide supporting evidence for potential causal relationships (Fig. 1B).

### Genetic variants serving as instruments
SNPs considered genetic instruments, were identified from published studies or MR-Base (Supplementary Data 2). For each SNP, the corresponding effect estimate on a trait expressed in *per* standard deviation (SD) units (assuming a *per* allele effect) and standard error (SE) was obtained. Only SNPs with a minor allele frequency >0.01 and a trait association of *P*-values $< 5 \times 10^{-8}$ in a European population GWAS were considered as instruments. We excluded correlated SNPs at a linkage

disequilibrium threshold of $r^2 > 0.01$, retaining SNPs with the strongest effect. For binary traits we restricted our analyses to traits with a medical diagnosis, excluding cancer. We removed duplicate exposure traits based on manual curation.

### Cancer GWAS summary statistics
To examine the association of each genetic instrument with cancer risk, we used summary GWAS effect estimates from: (1) Online consortia resources, for breast (BCAC; https://bcac.ccge.medschl.cam.ac.uk/, accessed July 2022) and prostate cancer (PRACTICAL; http://practical.icr.ac.uk/; accessed July 2022)[41,42]; (2) GWAS Catalog (https://www.ebi.ac.uk/gwas/), for ovarian, CRC, endometrial and lung cancers (accessed September 2022)[43–45]; (3) Investigators of published work, for RCC and oesophageal cancer[46–48]. Cancer subtype summary statistics were available for lung, breast, and ovarian cancers. As the UK Biobank was used to obtain genetic instruments for many traits investigated, the CRC and oesophageal GWAS association statistics were recalculated from primary data excluding UK Biobank samples to avoid sample overlap bias (Table 1). Single nucleotide polymorphisms were harmonised to ensure that the effect estimates of SNPs on

exposure traits and cancer risk referenced the same allele (Supplementary Data 12)[49].

## Statistical analysis

For each SNP, effects were estimated for cancer as an odds ratio (OR) per SD unit increase in the putative risk factor (ORSD), with 95% confidence intervals (CIs), using the Wald ratio[50]. For traits with multiple SNPs as IVs, causal effects were estimated under an inverse variance weighted random-effects (IVW-RE) model as the primary measurement as it is robust in the presence of pleiotropic effects, provided any heterogeneity is balanced at mean zero (Supplementary Data 3, 13-15)[51]. Weighted median estimate (WME) and mode-based estimates (MBE) were obtained to assess the robustness of findings (Supplementary Data 16)[52,53]. Directional pleiotropy was assessed using MR-Egger regression (Supplementary Data 17)[54]. The MR Steiger test was used to infer the direction of potentially causal effect for continuous exposure traits (Supplementary Data 18)[55]. For this we estimated the PVE using Cancer Research UK lifetime risk estimates for each tumour type (Supplementary Data 19). A leave-one-out strategy under the IVW-RE model was employed to assess the potential impact of outlying and pleiotropic SNPs (Supplementary Data 4)[56]. This sensitivity analysis tests the effect of performing MR on the IVs leaving one SNP out in turn. It can be used to identify when one SNP is driving the association as, when this SNP is removed, we can expect to see an attenuation of the MR association significance. Because two-sample MR of a binary risk factor and a binary outcome can be biased, we primarily considered whether there exists a significant non-zero effect, and only report ORs for consistency[57]. For proteomic traits which had an IV located *cis* (+/- 1 Mb) of the gene target we performed colocalisation using coloc[58]. This enumerates the four possible configurations of causal variants for two traits, calculating support for each model based on a Bayes factor. Adopting prior probabilities of $p_1$, $p_2 = 1 \times 10^{-4}$ and $p_{12} = 1 \times 10^{-5}$, a posterior probability $\geq 0.80$ was considered as supporting a specific model. For analyses of selected traits using MVMR we used the *mv_multiple* function in the TwoSampleMR package. MVMR was applied to investigate which of these traits within the same category had independent pleiotropic effects on a specific cancer. We restricted our MVMR analyses to traits which had $\geq 2$ IVs and for which we had access to full summary statistics required for the analysis. Statistical analyses were performed using the TwoSampleMR package v0.5.6 (https://github.com/MRCIEU/TwoSampleMR) and MendelianRandomization package in R (v3.4.0)[49].

## Estimation of study power

The power of MR to predict a causal relationship depends on the PVE by the instrument[59]. We excluded instruments with a F-statistic <10 since these are considered indicative of evidence for weak instrument bias[60]. We calculated conditional F-statistics for the traits using the *condFstat* function in the MendelianRandomzation package[61] (Supplementary Data 20). We estimated the genetic correlation between traits using Linkage-Disequilibrium Adjusted Kinships (LDAK) software (Supplementary Data 21). We derived LD matrices for the genetic variants using the *ld_matrix* function in TwoSampleMR. We estimated study power, stipulating a *P*-value of 0.05 for each target a priori across a range of effect sizes as *per* Brion et al. (Supplementary Data 2)[62]. Since power estimates for binary exposure traits and binary outcomes in a two-sample setting are unreliable, we did not estimate study power for binary traits[57].

## Assignment of statistical significance

The support for a causal relationship with non-binary traits was categorised into four hierarchical levels of statistical significance a priori: robust ($P_{\text{IVW-RE}} < 1.4 \times 10^{-5}$; corresponding to a *P*-value of 0.05 after Bonferroni correction for multiple testing (0.05/3,500), $P_{\text{WME}}$ or

$P_{\text{MBE}} < 0.05$, predicted true causal direction and >1 IVs), probable ($P_{\text{IVW-RE}} < 0.05$, $P_{\text{WME}}$ or $P_{\text{MBE}} < 0.05$, predicted true causal direction and >1 IVs), suggestive ($P_{\text{IVW-RE}} < 0.05$ or $P_{\text{WALD}} < 0.05$), and non-significant ($P_{\text{IVW-RE}} \geq 0.05$ or $P_{\text{WALD}} \geq 0.05$) (Supplementary Data 22). Robust associations are those that remain significant after correcting for multiple testing, the predicted direction of the effect is predicted to be from the exposure to the cancer risk and multiple MR methods report a significant association. We consider these associations to have the strongest statistical evidence, by virtue of the concordance between various MR methods and statistical validation tests. Probable associations are those that do not remain significant after correcting for multiple testing, but the remaining conditions are the same as for robust traits. We include this classification to account for the large number of traits tested in this analysis, noting that when taken in isolation these traits may be reported as having potentially causal associations with cancer. Suggestive traits are those in which show significance $P < 0.05$, but where one of the following conditions are flouted: the direction of effect may not be predicted to be from exposure to cancer outcome, or there is no significant consensus between the multiple MR methods. Additionally, significant associations for which only one SNP could be used as an IV are classified as suggestive. This was chosen to reflect the potential uncertainties that arise when performing MR using a Wald ratio test with a single IV. Finally, all other traits are classified as non-significant, indicating that it is unlikely that there is any potentially causal association. While non-significant associations can be due to low statistical power, they also indicate that a moderate causal effect is unlikely. For binary traits we classified associations as being supported ($P < 0.05$) or not supported ($P > 0.05$; Supplementary Data 6, 23-25).

## Support for causality

To strengthen evidence for causal relationships predicted from the MR analysis we exploited the semantic predications in Semantic MEDLINE Database (SemMedDB), which is based on all PubMed citations[63]. Within SemMedDB pairs of terms connected by a predicate which are collectively known as 'literature triples' (*i.e.* 'subject term 1' – predicates – 'object term 2'). These literature triples represent semantic relationships between biological entities derived from published literature. To interrogate SemMedDB we queried MELODI Presto and EpiGraphDB to facilitate data mining of epidemiological relationships for molecular and lifestyle traits[64–66]. For each putative risk factor-cancer pair the set of triples were overlapped, and common terms identified to reveal potentially causal pathways and inform aetiology. Based on the information profile of all literature mined triples, we considered literature spaces with >50 literature triples as being viable, corresponding to 90% of the information content[67]. We complemented this systematic text mining by referencing reports from the World Cancer Research Fund/American Institute for Cancer Research, and the International Agency for Cancer Research Global Cancer Observatory, as well as querying specific putative relationships in PubMed[7].

# Data availability

Genetic instruments can be obtained through MR-Base or from published work (Supplementary Data 2). Summary GWAS cancer data are available from: https://bcac.ccge.medschl.cam.ac.uk/bcacdata/ (breast cancer); http://practical.icr.ac.uk/blog/?page_id=8088 (prostate cancer); GWAS Catalogue ID: GCST004481 (ovarian cancer); GWAS Catalogue ID: GCST006464 (endometrial cancer); GWAS Catalogue ID: GCST004748 (lung cancer); direct communication with consortia (renal and esophageal cancers); - phs001415.v1.p1, phs001315.v1.p1, phs001078.v1.p1, phs001903.v1.p1, phs001856.v1.p1 and phs001045.v1.p1 (US based studies) and GWAS Catalog ID: GCST90129505 (European based studies) colorectal cancer. Source data are provided within the supplementary data of this paper.

## Code availability

We provide custom code used to generate the results presented in this study at https://github.com/houstonlab/MR-PheWAS

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

## Acknowledgements

R.S.H. acknowledges grant support from Cancer Research UK (C1298/A8362), the Wellcome Trust (214388) and Myeloma UK. A.S. is in receipt of a National Institute for Health Research (NIHR) Academic Clinical Lectureship, funding from the Royal Marsden Biomedical Research Centre, a Starter Grant from the Academy of Medical Sciences and is the recipient of a Wellcome Trust Early Career Award (227000/Z/23/Z). M.K. is supported by a fellowship from the David Forbes-Nixon Foundation. We acknowledge pump-priming funding from the Royal Marsden Biomedical Research Centre Early Diagnosis, Detection and Stratified Prevention Theme. This is a summary of independent research supported by the NIHR Biomedical Research Centre at the Royal Marsden NHS Foundation Trust and the Institute of Cancer Research. The views expressed are those of the author(s) and not necessarily those of the NHS, the NIHR or the Department of Health. Support from the DJ Fielding Medical Research Trust is also acknowledged. A.H. was in receipt of a summer studentship from the Genetics Society. We thank Alex Cornish for providing code and critically appraising the manuscript. The breast cancer genome-wide association analyses for BCAC and CIMBA were supported by Cancer Research UK (PPRPGM-Nov20\100002, C1287/A10118, C1287/A16563, C1287/A10710, C12292/A20861, C12292/A11174, C1281/A12014, C5047/A8384, C5047/A15007, C5047/A10692, C8197/A16565) and the Gray Foundation, The National Institutes of Health (CA128978, X01HG007492- the DRIVE consortium), the PERSPECTIVE project supported by the Government of Canada through Genome Canada and the Canadian Institutes of Health Research (grant GPH-129344) and the Ministère de l'Économie, Science et Innovation du Québec through Genome Québec and the PSRSIIRI-701 grant, the Quebec Breast Cancer Foundation, the European Community's Seventh Framework Programme under grant agreement n° 223175 (HEALTH-F2-2009-223175) (COGS), the European Union's Horizon 2020 Research and Innovation Programme (634935 and 633784), the Post-Cancer GWAS initiative (U19 CA148537, CA148065 and CA148112 - the GAME-ON initiative), the Department of Defence (W81XWH-10-1-0341), the Canadian Institutes of Health Research (CIHR) for the CIHR Team in Familial Risks of Breast Cancer (CRN-87521), the Komen Foundation for the Cure, the Breast Cancer Research Foundation and the Ovarian Cancer Research Fund. All studies and funders are listed in Zhang H et al. (Nat Genet, 2020). The colorectal cancer genome-wide association analysis was supported by Ulrike Peters (GECCO), Stephanie Schmit (CCFR), Stephen Gruber (CORECT), Ian Tomlinson (CORGI, SCOT), and Malcolm Dunlop (SOCCS). Full study details and funders are listed in Fernandez-Rozadilla C et al. (Nat Genet, 2023). The Prostate cancer genome-wide association analyses are supported by the Canadian Institutes of Health Research, European Commission's Seventh Framework Programme grant agreement n° 223175 (HEALTH-F2-2009-223175), Cancer Research UK Grants C5047/A7357, C1287/A10118, C1287/A16563, C5047/A3354, C5047/A10692, C16913/A6135, and The National Institute of Health (NIH) Cancer Post-Cancer GWAS initiative grant: No. 1 U19 CA 148537-01 (the GAME-ON initiative). We would also like to thank the following for funding support: The Institute of Cancer Research and The Everyman Campaign, The Prostate Cancer Research Foundation, Prostate Research Campaign UK (now PCUK), The Orchid

Cancer Appeal, Rosetrees Trust, The National Cancer Research Network UK, The National Cancer Research Institute (NCRI) UK. We are grateful for support of NIHR funding to the NIHR Biomedical Research Centre at The Institute of Cancer Research and The Royal Marsden NHS Foundation Trust. The Prostate Cancer Program of Cancer Council Victoria also acknowledge grant support from The National Health and Medical Research Council, Australia (126402, 209057, 251533, 396414, 450104, 504700, 504702, 504715, 623204, 940394, 614296,), VicHealth, Cancer Council Victoria, The Prostate Cancer Foundation of Australia, The Whitten Foundation, PricewaterhouseCoopers, and Tattersall's. EAO, DMK, and EMK acknowledge the Intramural Program of the National Human Genome Research Institute for their support. Genotyping of the OncoArray was funded by the US National Institutes of Health (NIH) [U19 CA 148537 for ELucidating Loci Involved in Prostate cancer SuscEptibility (ELLIPSE) project and X01HG007492 to the Center for Inherited Disease Research (CIDR) under contract number HHSN268201200008I] and by Cancer Research UK grant A8197/A16565. Additional analytic support was provided by NIH NCI U01 CA188392 (PI: Schumacher). Funding for the iCOGS infrastructure came from: the European Community's Seventh Framework Programme under grant agreement n° 223175 (HEALTH-F2-2009-223175) (COGS), Cancer Research UK (C1287/A10118, C1287/A 10710, C12292/A11174, C1281/A12014, C5047/A8384, C5047/A15007, C5047/A10692, C8197/A16565), the National Institutes of Health (CA128978) and Post-Cancer GWAS initiative (1U19 CA148537, 1U19 CA148065 and 1U19 CA148112 – the GAME-ON initiative), the Department of Defence (W81XWH-10-1-0341), the Canadian Institutes of Health Research (CIHR) for the CIHR Team in Familial Risks of Breast Cancer, Komen Foundation for the Cure, the Breast Cancer Research Foundation, and the Ovarian Cancer Research Fund. The BPC3 was supported by the U.S. National Institutes of Health, National Cancer Institute (cooperative agreements U01-CA98233 to D.J.H., U01-CA98710 to S.M.G., U01-CA98216 toE.R., and U01-CA98758 to B.E.H., and Intramural Research Program of NIH/National Cancer Institute, Division of Cancer Epidemiology and Genetics). CAPS GWAS study was supported by the Swedish Cancer Foundation (grant no 09-0677, 11-484, 12-823), the Cancer Risk Prediction Center (CRisP; www.crispcenter.org), a Linneus Centre (Contract ID 70867902) financed by the Swedish Research Council, Swedish Research Council (grant no K2010-70X-20430-04-3, 2014-2269). PEGASUS was supported by the Intramural Research Program, Division of Cancer Epidemiology and Genetics, National Cancer Institute, National Institutes of Health.

## Author contributions

Contribution: M.W., A.S., C.M. and R.S.H. designed the study. M.W., A.S., C.M., A.H., R.C., J.V. and P.L. performed statistical analyses; I.G., Ca.M., J.S., and C.P. performed genome-wide association studies of cancers included in the study; M.W, A.S., C.M., M.K. and R.S.H. drafted the manuscript; all authors reviewed, read, and approved the final manuscript.

## Competing interests

The authors declare no competing interests.

### Ethics approval

The analysis was undertaken using published GWAS data, hence ethical approval was not required.
