## [Peer Review File · Nature Communications]

Phenome-wide Mendelian randomisation analysis of 378,142 cases reveals risk factors for eight common cancers

Phenome-wide
Mendelian
randomisation analysis
of 378,142 cases
reveals risk
2 factors for eight
common cancersREVIEWER COMMENTS

Reviewer #1 (Remarks to the Author): expertise in epidemiology statistics and Mendelian randomisation

The authors have conducted a large scale MR Phewas analysis of more than 3,000 exposures on 8 cancers, and created an app for showing these associations. I want to congratulate them for such a huge effort, which helps map a large number of MR associations. They also attempted to judge the potential causality of these associations by using a few MR analyses (IVW RE, weighted median, mode, Steiger filtering), and supported them with some literature text mining approaches. My main comment for this "automatic" approach of judging the potential for causality for thousands of associations is that it is difficult to make a judgement without being able to go deeper and examine the function of each IV. There are several examples where additional analyses would be needed to create more certainty e.g.

1) There were several probable and suggestive associations for dietary intakes and cancer risk (i.e. fish, cheese, beef, poultry), where a closer look at the functionality and potential pleiotropy of the used IVs would be warranted and analyses supported by MVMR analyses adjusting for potential pleiotropic pathways.

2) For lipids/lipoproteins, we know that they are mutually correlated and MVMR analyses are needed to clarify their independent associations.

3) Fatty acids are very pleiotropic and correlated traits and more careful analyses are also needed here.

4) For associations of plasma proteomic markers and other biomarkers where a clear gene target exists, analyses using cis IV selection and colocalization analyses are warranted.

A few more minor comments:

1) Report direction of association for paragraph describing associations with dietary intakes

lines 125-132

2) First paragraph of Discussion, What % of associations have not been previously assessed by MR, and what % of associations that found some evidence of association in this paper have not been previously assessed by MR

3) There is some lack of clarity in the criterion used for judging causality using the number of SNPs: >1 IV mentioned in Methods text

vs. on sup table 17, association not driven by single SNP

Reviewer #2 (Remarks to the Author): expertise in pan cancer GWAS studies

This study conducted a Mendelian randomization phenome-wide association study (MR-PheWAS) using summary data from genome-wide association studies (GWAS) to identify causal relationships for eight common cancers. They identified established and novel risk factors for cancer, and provided a web app for visualizing the results.

However, the manuscript has limited content and lacks sufficient depth. As a result, the paper does not present impactful findings and fails to convincingly convey the importance and relevance of the research. Additionally, the figures in this article are simplistic and do not effectively support the presentation of the content. In my opinion, the overall quality of the paper does not meet the publication standards of Nature Communications.

There are some suggestions that can improve the quality of the manuscript:

1. The web app serves as a valuable resource for cancer research. It is suggested that the authors assign a suitable name to the web app. Additionally, the authors can include a dedicated section in the results to introduce the functionality of the web app and provide guidance on its utilization for in-depth exploration and analysis.
2. Line 86-90: authors should provide relevant references to support categories.
3. Line 439: authors should expand the legend for Figure 1B.
4. Line 454: authors should provide more detail on the legend for Figure 4.
5. Figure 4 is too small to read and should be enlarged for better visibility and clarity.

6. Figure 5 could benefit from performing row clustering within each class to enhance the visualization of the effect size distribution

7. The authors should consider sharing the analysis code to promote learning and communication in this field.

Reviewer #1

The authors have conducted a large-scale MR Phewas analysis of more than 3,000 exposures on 8 cancers and created an app for showing these associations. I want to congratulate them for such a huge effort, which helps map a large number of MR associations. They also attempted to judge the potential causality of these associations by using a few MR analyses (IVW RE, weighted median, mode, Steiger filtering), and supported them with some literature text mining approaches. My main comment for this "automatic" approach of judging the potential for causality for thousands of associations is that it is difficult to make a judgement without being able to go deeper and examine the function of each IV. There are several examples where additional analyses would be needed to create more certainty e.g.

- 1.1) There were several probable and suggestive associations for dietary intakes and cancer risk (i.e. fish, cheese, beef, poultry), where a closer look at the functionality and potential pleiotropy of the used IVs would be warranted and analyses supported by MVMR analyses adjusting for potential pleiotropic pathways.
- 1.2) For lipids/lipoproteins, we know that they are mutually correlated and MVMR analyses are needed to clarify their independent associations.
- 1.3) Fatty acids are very pleiotropic and correlated traits and more careful analyses are also needed here.

Response: We acknowledge these points and, as suggested, now include the results from multivariable mendelian randomisation (MVMR) for dietary intake, lipids/lipoprotein and fatty acids. We also include the results of MVMR as a form of mediation analysis for selected traits identified after systematic mining of the literature .

- 1.4) For associations of plasma proteomic markers and other biomarkers where a clear gene target exists, analyses using cis IV selection and colocalization analyses are warranted.

Response: As requested we have updated our manuscript to include Bayesian co-localisation for proteomic markers with an IV *cis* to the gene target; specifically, filtering to include those IVs which showed a highly significant association with cancer ($P < 1 \times 10^{-6}$).

- 1.5) Report direction of association for paragraph describing associations with dietary intakes lines 125-132

Response: We have modified the text to include direction of effect between the trait and the outcome as follows:

“With respect to dietary intake our analysis demonstrated probable associations between genetically predicted higher levels of coffee, oily fish, and cheese intake with reduced CRC risk and suggestive associations between genetically predicted beef and poultry intake and elevated CRC risk. We found suggestive associations between genetically predicted high serum vitamin B12 with increased risk of colorectal and prostate cancer, higher serum calcium and 25-hydroxyvitamin-D with increased risk of RCC, higher low blood selenium with decreased risk of colorectal and oesophageal cancers and higher

methionine and zinc with reduced CRC risk. We observed no association between genetically predicted blood levels of circulating carotenoids or vitamins B6 and E for any of the cancers.”

1.6) First paragraph of Discussion, What % of associations have not been previously assessed by MR, and what % of associations that found some evidence of association in this paper have not been previously assessed by MR.

Response: Comparing our analysis to the recent systematic review MR studies conducted by Markozannes *et al* (which we now reference), we estimate that less than 10% of the MR exposures in this study had been the subject of previous investigations. In addition, 85% of those traits which were significant had not previously been examined. We have updated the manuscript to provide this information.

1.7) There is some lack of clarity in the criterion used for judging causality using the number of SNPs: >1 IV mentioned in Methods text vs. on sup table 17, association not driven by single SNP

Response: To avoid ambiguity we have rephrased the wording in the supplementary table to indicate what is meant by “Association not driven by a single SNP”. We have replaced this criterion with “For traits with multiple IVs, the association is not driven by single SNP as indicated by leave-one-out test”. We have also clarified in the methods what the leave-one-out test is used for: “This sensitivity analysis tests the effect of performing MR on the IVs leaving one SNP out in turn. It can be used to identify when one SNP is driving the association as, when this SNP is removed, we can expect to see an attenuation of the MR association significance”.

Reviewer #2

This study conducted a Mendelian randomization phenome-wide association study (MR-PheWAS) using summary data from genome-wide association studies (GWAS) to identify causal relationships for eight common cancers. They identified established and novel risk factors for cancer and provided a web app for visualizing the results.

However, the manuscript has limited content and lacks sufficient depth. As a result, the paper does not present impactful findings and fails to convincingly convey the importance and relevance of the research. Additionally, the figures in this article are simplistic and do not effectively support the presentation of the content. In my opinion, the overall quality of the paper does not meet the publication standards of Nature Communications.

There are some suggestions that can improve the quality of the manuscript:

2.1) The web app serves as a valuable resource for cancer research. It is suggested that the authors assign a suitable name to the web app. Additionally, the authors can include a dedicated section in the results to introduce the functionality of the web app and provide guidance on its utilization for in-depth exploration and analysis.

Response: We now include a landing page on the app, which introduces the resource and provides comprehensive guidance to the user. The landing page can be accessed at any given time and includes visual annotations of the app. We have introduced the app in the results section of the manuscript. The name of the app “Mendelian Randomisation of Cancers (MRCan)” is more prominently defined.

2.2) Line 86-90: authors should provide relevant references to support categories.

Response: We have included reference to a review of MR studies which inspired the choice of categories. We have clarified this in the manuscript:

“To aid interpretation we grouped traits ... into the following categories, using a similar approach to Markozannes *et al*: cardiometabolic; dietary intake; anthropometrics; immune and inflammatory factors; fatty acid (FA) and lipoprotein metabolism; lifestyle, reproduction, education and behaviour; metabolomics and proteomics; miscellaneous.”

2.3) Line 439: authors should expand the legend for Figure 1B.

Response: As requested we have now included a descriptive legend for Figure 1B. Specifically, “Genetic variants serving as instruments for exposure traits under investigation were identified from MR Base or PubMed. GWAS data for the eight cancers was acquired and MR analysis was performed. Results were triangulated through literature mining to provide supporting evidence for causal relationships”.

2.4) Line 454: authors should provide more detail on the legend for Figure 4.

Response: As requested we now include additional information in the legend for Figure 4. Specifically:

“These diagrams illustrate the relationship between trait and outcome via their linked literature triples. The thickness of the line connecting two traits indicates the frequency with which that triple is mentioned in the literature. Relationships for: (a) *IGF-1* and colorectal cancer; (b) *IL-18* and lung cancer; (c) *LAG-3* and endometrial cancer; (d) *PRDX1* and prostate cancer.”

2.5) Figure 4 is too small to read and should be enlarged for better visibility and clarity.

Response: We apologise for the lack of clarity and have now provided a revised image at 300ppi resolution.

2.6) Figure 5 could benefit from performing row clustering within each class to enhance the visualization of the effect size distribution.

Response: We acknowledge this point and have revised Figure 5 accordingly.

2.7) The authors should consider sharing the analysis code to promote learning and communication in this field.

Response: As requested we now provide analysis code supplementary to that which is already publicly available, and we have updated the text in the data availability section to reflect this.

REVIEWERS' COMMENTS:

Reviewer #1 (Remarks to the Author):

The authors seem to have addressed this reviewer comments, but unfortunately I couldn't find a version of the paper with tracked changes to assist in identifying the areas of the manuscript where changes have been applied. Please provide such a document. In addition, the authors have now added several MVMR analyses for dietary variables, lipids and fatty acids. Even though these analyses are towards the right direction, there are issues that need improvement.

1. It is not clear how these models were built, as the reply to the reviewer comment was very short. I assume that the authors put in the MVMR models (related; how decided?) variables that were nominally significant in the univariable MR (UVMR) analyses, but the approach needs to be clarified.
2. This "automatic" selection of variables creates issues, namely inclusion of variables with small number of IVs (even 1). The authors should estimate the conditional F statistics and remove variables with conditional F lower than 10, as their inclusion will bias the MVMR findings. In addition, sometimes variables that are very highly correlated (like apoA and HDL) end up in the same model, but MVMR cannot handle correlations larger than 90%, as they are basically the same variable.
3. I would suggest that the authors build MVMR models for significant dietary variables, lipids and fatty acids in a more knowledgeable fashion focusing on each finding separately and taking into consideration information on potential pleiotropy from current knowledge (i.e. existing GWAS studies). For example, if a certain lipid e.g. LDL has significant UVMR results, a MVMR model should be built including other lipoprotein components, like LDL, HDL, Tg, L(a), etc. regardless if the other lipoproteins have significant or insignificant results. Incorporating total cholesterol in this model doesn't make sense, as it would be correlated with the sum of all the components.

For the above reasons, I consider the MVMR results not interpretable as currently

presented.

Reviewer #2 (Remarks to the Author):

It is surprisingly that the authors did not response to the previous major comments at all. "However, the manuscript has limited content and lacks sufficient depth. As a result, the paper does not present impactful findings and fails to convincingly convey the importance and relevance of the research. Additionally, the figures in this article are simplistic and do not effectively support the presentation of the content. In my opinion, the overall quality of the paper does not meet the publication standards of Nature Communications."

Reviewer #1 (Remarks to the Author):

The authors seem to have addressed this reviewer comments, but unfortunately I couldn't find a version of the paper with tracked changes to assist in identifying the areas of the manuscript where changes have been applied. Please provide such a document. In addition, the authors have now added several MVMR analyses for dietary variables, lipids and fatty acids. Even though these analyses are towards the right direction, there are issues that need improvement.

Response: We thank the reviewer for acknowledging we have addressed the reviewer's comments. We submitted a "tracked changes" version with our resubmission. We would be happy to resend this.

1.It is not clear how these models were built, as the reply to the reviewer comment was very short. I assume that the authors put in the MVMR models (related; how decided?) variables that were nominally significant in the univariable MR (UVMR) analyses, but the approach needs to be clarified.

Response: We specifically state in the revised text: "We restricted our MVMR analyses to traits which had an association $P < 0.05$, ≥ 2 IVs and which we had access to full summary statistics required for the analysis".

2.This "automatic" selection of variables creates issues, namely inclusion of variables with small number of IVs (even 1). The authors should estimate the conditional F statistics and remove variables with conditional F lower than 10, as their inclusion will bias the MVMR findings. In addition, sometimes variables that are very highly correlated (like apoA and HDL) end up in the same model, but MVMR cannot handle correlations larger than 90%, as they are basically the same variable.

Response: As stated in response to comment 1 we required two or more IVs for each trait. Where the number of traits is greater than the total number of IVs, MVMR cannot be performed; we have ensured this never happens in our work. Inevitably, harmonising multiple IVs across traits will result in the number of IVs for each trait reducing, in some cases to one. We chose two or more IVs to ensure this resulted in at least one IV for each trait included in the models. Furthermore, we are happy to provide the conditional F statistics (>10) for completeness.

3. I would suggest that the authors build MVMR models for significant dietary variables, lipids and fatty acids in a more knowledgeable fashion focusing on each finding separately and taking into consideration information on potential pleiotropy from current knowledge (i.e. existing GWAS studies). For example, if a certain lipid e.g. LDL has significant UVMR results, a MVMR model should be built including other lipoprotein components, like LDL, HDL, Tg, L(a), etc. regardless if the other lipoproteins have significant or insignificant results. Incorporating total cholesterol in this model doesn't make sense, as it would be correlated with the sum of all the components.

For the above reasons, I consider the MVMR results not interpretable as currently presented.

Response: The approach of our MRPhEWAS is to agnostically assess causal relationships between >3,000 traits and eight cancer types. As the reviewer initially requested, we have extended this agnostic approach by using the results of the MRPhEWAS to inform the MVMR. We assert that utilising existing knowledge as the reviewer now suggests is counter intuitive and negates any advantage of MRPhEWAS.

Regarding Reviewer 2 comments:

Reviewer #2 (Remarks to the Author)

It is surprisingly that the authors did not response to the previous major comments at all. "However, the manuscript has limited content and lacks sufficient depth. As a result, the paper does not present impactful findings and fails to convincingly convey the importance and relevance of the research. Additionally, the figures in this article are simplistic and do not effectively support the presentation of the content. In my opinion, the overall quality of the paper does not meet the publication standards of Nature Communications."

Response: Since our first submission we have re-written the manuscript and performed a number of new analyses including: (i) multivariable MR (MVMR); (ii) cis IV selection and colocalisation analysis for plasma proteomic markers; (iii) redesigning the web application; (iv) modifying the text and figures in response to specific suggestions from both reviewers; (v) shared our analysis code. We thank the reviewers for providing specific feedback which has improved our

manuscript. We therefore find it surprising that reviewer 2 concludes we have not responded to any of the major comments when objectively we have responded to all comments. We would be happy to revise the text or figures to address any other specific concerns.

REVIEWER COMMENTS

Reviewer #3 (Remarks to the Author):

The authors have performed a large MR PheWAS for eight cancer types including over 378K cases and 485K controls including over 3,600 risk factors. This is an impressive amount of work with the aim to agnostically investigate relationships between multiple traits and risk of different cancer types.

Reviewer 1: Authors have replied to points 1.1 – 1.6.

Reviewer 2: Authors seem to have answered points 2.1 through 2.7 from the first round of reviews that R1 listed as changes that “can improve the quality of the manuscript”. However, it appears that the reviewer was also expecting them to respond to the comment in the first paragraph: “However, the manuscript has limited content and lacks sufficient depth. As a result, the paper does not present impactful findings and fails to convincingly convey the importance and relevance of the research. Additionally, the figures in this article are simplistic and do not effectively support the presentation of the content. In my opinion, the overall quality of the paper does not meet the publication standards of Nature Communications.”

While this manuscript describes an immense amount of work there are important improvements that need to be made. The main criticisms are listed below.

1. Using the word “causal” for associations throughout the manuscript comes across as if the authors are conveying results as facts. Without RCT as the gold standard to formally establish causality, such wording is not quite appropriate. Examples are on page 5: “Many of the identified causal relationships, especially ...”. Another example is on page 10 (discussion second paragraph): “A number of the causal relationships we identified ...”. These (and many other) statements are too strong and should be toned down using “possibly causal” or “potentially causal” or similar language. This holds particularly true for the multiple probable and suggestive associations that are described throughout the manuscript. Also note that only robust associations should be mentioned in the abstract.

2. The grouping of robust, probable, and suggestive associations appears somewhat arbitrary, and it is not clear if the latter two classes should be highlighted in the manuscript as much as what is currently done. The authors should describe better the rationale behind their probable and suggestive criteria and why such associations could, or could not, represent valid findings. Also, betas and P values should be listed in the results section so that the reader can more easily distinguish between associations that are clearly significant to those near the threshold of significance. Finally, in the discussion it is unclear which associations are being described that are probable and suggestive as compared to robust. This is quite confusing.

3. Coronary Artery Disease appears to have robust associations for many of the cancers (see Table S6). Why is this not mentioned in the results or discussion?

4. The authors should provide F statistics estimates after removing variables with conditional F lower than 10 (as per reviewer 1). Furthermore, I do agree that the MVRM model should incorporate current knowledge (i.e., as per the LDL example from reviewer 1, Q3 second round). It is commendable that the authors have put in all this work to examine all these associations in an agnostic manner, but when the literature holds important information, this needs to be taken into account.

5. Figures and tables still need more detailed legends than what is presented in the tracked version. This is especially needed for supplementary tables.

Response to reviewers

Reviewer #3 (Remarks to the Author):

The authors have performed a large MR PheWAS for eight cancer types including over 378K cases and 485K controls including over 3,600 risk factors. This is an impressive amount of work with the aim to agnostically investigate relationships between multiple traits and risk of different cancer types.

Reviewer 1: Authors have replied to points 1.1 – 1.6.

Reviewer 2: Authors seem to have answered points 2.1 through 2.7 from the first round of reviews that R1 listed as changes that “can improve the quality of the manuscript”. However, it appears that the reviewer was also expecting them to respond to the comment in the first paragraph: “However, the manuscript has limited content and lacks sufficient depth. As a result, the paper does not present impactful findings and fails to convincingly convey the importance and relevance of the research. Additionally, the figures in this article are simplistic and do not effectively support the presentation of the content. In my opinion, the overall quality of the paper does not meet the publication standards of Nature Communications.”

While this manuscript describes an immense amount of work there are important improvements that need to be made. The main criticisms are listed below.

1. Using the word “causal” for associations throughout the manuscript comes across as if the authors are conveying results as facts. Without RCT as the gold standard to formally establish causality, such wording is not quite appropriate. Examples are on page 5: “Many of the identified causal relationships, especially ...”. Another example is on page 10 (discussion second paragraph): “A number of the causal relationships we identified ...”. These (and many other) statements are too strong and should be toned down using “possibly causal” or “potentially causal” or similar language. This holds particularly true for the multiple probable and suggestive associations that are described throughout the manuscript. Also note that only robust associations should be mentioned in the abstract.

Response: We thank the reviewer for the constructive review of our manuscript. In light of this specific comment, we have edited the manuscript text and rather than stating “causal associations” we now state “potentially causal associations”. We have also revised the main text to state that we have “predicted”, rather than “identified”, causal associations using MR.

With respect to the Abstract, we acknowledge our original text summarizing findings might be misinterpreted and have revised it accordingly. Specifically, we now state “In addition to

providing supporting evidence for well-established risk factors (smoking, alcohol, obesity, lack of physical activity), we also implicate specific factors, including sex steroid hormones, plasma lipids, and telomere length as determinants of cancer risk.”

2. The grouping of robust, probable, and suggestive associations appears somewhat arbitrary, and it is not clear if the latter two classes should be highlighted in the manuscript as much as what is currently done. The authors should describe better the rationale behind their probable and suggestive criteria and why such associations could, or could not, represent valid findings. Also, betas and P values should be listed in the results section so that the reader can more easily distinguish between associations that are clearly significant to those near the threshold of significance. Finally, in the discussion it is unclear which associations are being described that are probable and suggestive as compared to robust. This is quite confusing.

Response: We now include an additional Figure detailing the statistical criteria for assigning grouping of robust, probable, and suggestive associations within the main text to complement what is described in an expanded Methods section. We have also updated the results section to explicitly report odds ratios and associated P-values. Finally, we have revised the discussion to include the statistical classification of highlighted results to further aid interpretation.

3. Coronary Artery Disease appears to have robust associations for many of the cancers (see Table S6). Why is this not mentioned in the results or discussion?

Response: Coronary Artery Disease (CAD) is a binary trait and as specified in the manuscript “two-sample MR of a binary risk factor and a binary outcome can be biased, we primarily considered whether there exists a significant non-zero effect, and only report ORs for consistency” (Burgess & Labrecque, 2018). This issue is explicitly stated within the text. Hence, we therefore do not assign a statistical classification (Robust, Probable, Suggestive) to binary traits and this is reflected in the data presented in Table S6. The reviewer is correct in noting that an association with CAD is *supported* for many cancers, however we choose not to discuss this further due to the potential biases per outlined in the reference above.

4. The authors should provide F statistics estimates after removing variables with conditional F lower than 10 (as per reviewer 1). Furthermore, I do agree that the MVRM model should incorporate current knowledge (i.e., as per the LDL example from reviewer 1, Q3 second round). It is commendable that the authors have put in all this work to examine all these associations in

an agnostic manner, but when the literature holds important information, this needs to be taken into account.

Response: We have amended our work on MVMR to incorporate relevant examples from the literature, as suggested. We have calculated conditional F-statistics for traits included in MVMR analysis. While calculating conditional F-statistics for dietary traits we noted weak instrument bias, which precludes their inclusion in a MVMR model. We note this in discussion. While lack of sufficient IVs and availability of summary level data precludes MVMR of fatty acids, we have commented on the previously observed relationship within the omega-3 and omega-6 fatty acids, specifically referring to the leave one out and single SNP analysis performed in the manuscript, which implicates a shared SNP at the FADS locus as driving the majority of these associations. We have also made code available should readers wish to pursue further MVMR.

5. Figures and tables still need more detailed legends than what is presented in the tracked version. This is especially needed for supplementary tables.

Response: We now provide more extensive legends for the supplementary tables. We also added more detail to figure legends where appropriate.

REVIEWERS' COMMENTS

Reviewer #3 (Remarks to the Author):

The authors have answered all my questions.